lncRNA MALAT1 mediates osteogenic differentiation of bone mesenchymal stem cells by sponging miR-129-5p

Yin Junhao 1 2
Zheng Zhanglong 1 2
Zeng Xiaoli 1 2
Zhao Yijie 2 3
Ai Zexin 1 2
Yu Miao 1 2
Wu Yang’ou 2 4
Jiang Jirui 1 2
Li Jia lj2014@tongji.edu.cn 2 3
Li Shengjiao 07824@tongji.edu.cn 1 2
1 Department of Oral and Maxillofacial Surgery, School and Hospital of Stomatology, Tongji University , Shanghai , China
2 Shanghai Engineering Research Center of Tooth Restoration and Regeneration , Shanghai , China
3 Department of Prosthodontics, School and Hospital of Stomatology, Tongji University , Shanghai , China
4 Department of Oral and Maxillofacial Surgery, Shanghai Xuhui District Dental Center, Jiaotong University , Shanghai , China
Ogunwobi Olorunseun
Electronic publication date: 2022 Apr 22
Publication date: 2022
Volume: 10
Electronic Location ID: e13355
Received 2022 Jan 25; Accepted 2022 Apr 8
Copyright: ©2022 Yin et al.
Copyright year: 2022
Copyright holder: Yin et al.
License: This is an open access article distributed under the terms of the Creative Commons Attribution License, which permits unrestricted use, distribution, reproduction and adaptation in any medium and for any purpose provided that it is properly attributed. For attribution, the original author(s), title, publication source (PeerJ) and either DOI or URL of the article must be cited.
License URL: https://creativecommons.org/licenses/by/4.0/

Keywords: MALAT1, miR-129-5p, Bone mesenchymal stem cells, Osteoblast differentiation

Funding: Shanghai Municipal Natural Science Foundation 19140904800 22ZR1467200 This work was supported by the Shanghai Municipal Natural Science Foundation (No. 22ZR1467200 and No. 19140904800). The funders had no role in study design, data collection and analysis, decision to publish, or preparation of the manuscript.

==============================
Background

Bone mesenchymal stem cells (BMSCs) have good osteogenic differentiation potential and have become ideal seed cells in bone tissue engineering. However, the osteogenic differentiation ability of BMSCs gradually weakens with age, and the regulatory mechanism is unclear.

Method

We conducted a bioinformatics analysis, dual-luciferase reporter (DLR) experiment, and RNA binding protein immunoprecipitation (RIP) to explore the hub genes that may affect BMSC functions.

Results

The expression level of long non-coding RNA (lncRNA) metastasis-associated lung adenocarcinoma transcript 1 (Malat1) was significantly higher in the BMSCs from elderly than younger mice, while miR-129-5p showed the opposite trend. The results of alkaline phosphatase staining, quantitative reverse transcription PCR and western blot experiments indicated that inhibiting the expression of Malat1 inhibits the osteogenic differentiation of BMSCs. This effect can be reversed by reducing the expression of miR-129-5p. Additionally, DLR and RIP experiments confirmed that Malat1 acts as a sponge for miR-129-5p.

Conclusion

Overall, our study findings indicated that lncRNA Malat1 may play a critical role in maintaining the osteoblast differentiation potential of BMSCs by sponging miR-129-5p.

Introduction

Bone mesenchymal stem cells (BMSCs) are widely studied multipotent stem cells found in bone marrow. Their strong plasticity and differentiation potential make them useful for tissue repair (Chu et al., 2020; Fu et al., 2019). During the aging process, the osteogenic differentiation ability of BMSCs gradually weakens, resulting in a decrease in osteoblasts and the accumulation of adipocytes, eventually leading to diseases such as senile osteoporosis. Several key factors have been identified in the osteogenesis and adipogenesis of BMSCs. Peroxisome proliferator-activated receptor gama (PPARG) and core-binding factor alpha1 (CEBPA) are the most important regulators of BMSC adipogenic differentiation. Runt-related transcription factor (RUNX2) and Osterix play essential roles during osteogenic differentiation of BMSCs (Qadir et al., 2020). Although many studies have investigated the differentiation fate of BMSCs, the exact mechanism is still unknown.

Long non-coding RNA (lncRNA) is a type of non-coding RNA longer than 200 nucleotides. With the continuous development of high-throughput sequencing technology, a growing number of studies have shown that lncRNA participates in gene expression regulation and cell differentiation (Huynh et al., 2017). For example, Zhuang et al. (2015) found that lncRNA MEG3 promotes osteogenic differentiation of BMSCs in patients with multiple myeloma by targeting the transcription of bone morphogenetic protein. LncRNAs are widely involved in chromosome silencing, chromatin modification, transcriptional activation, and transcriptional interference. Among these, competitive endogenous RNA (ceRNA) is one of the main mechanisms of lncRNA regulation. CeRNAs have miRNA response elements that enable them to bind target miRNAs, thereby regulating their quantity and function. For instance, according to the findings of Yang et al. (2019a), lncRNA ORLNC1 functions as a ceRNA for miR-296 and regulates the balance between osteogenesis and adipogenesis of BMSCs. The study also found that overexpression of ORLNC1 led to decreases in the amount and density of trabecular bone in mouse femurs (Yang et al., 2019a). MicroRNA, another representative non-coding RNA of 18–25 nucleotides in length (Bartel, 2009), is also involved in Wnt/β-catenin and TGF-β/BMP/Smad signaling pathways to control osteogenesis (Zhang et al., 2021).

This current study used bioinformatics analysis to investigate the differential expressed non-coding RNAs in BMSCs from middle-aged and elderly adults, finding lncRNA Malat1 highly expressed in the elderly group. Previous research found that Malat1 functioned as a miRNA decoy for miR-30 and thus influenced the cell fate determination of adipose-derived mesenchymal stem cells (Yi, Liu & Xiao, 2019). Furthermore, Malat1 was also determined to promote osteogenesis differentiation and possibly act as an inhibitory regulator in steroid-induced avascular necrosis of the femoral head by sponging miR-214 (10). However, the role of Malat1 in regulating the osteogenic differentiation of BMSCs is largely unknown. In this research, we found that Malat1 promoted the osteogenic differentiation of BMSCs, and its function was related to miR-129-5p. These results contribute to a better understanding of the possible mechanism of stem cell differentiation and advance the use of stem cells in tissue engineering.

Materials and Methods

GEO data collection

Human expression profiles (accession number: GSE35955) were downloaded from the NCBI GEO (Edgar, 2002; Oghumu et al., 2016), including the lncRNA and mRNA expression data of BMSCs derived from five middle-aged (42–67 year old) and four elderly (78–89 year old) donors. The data were then analyzed using the GPL570 Affymetrix Human Genome U133 Plus 2.0 Array. The associated web link is displayed in File S1.

Data quality control

The quality of the nine samples was investigated using the R package AffyPLM (Heber & Sick, 2006). The boxplot representations of the relative log expression and normalized unscaled standard errors are two commonly used chip quality control methods.

Data pre-processing

The transcripts were divided into two categories (1,205 lncRNAs and 16,418 mRNAs), utilizing the comprehensive gene annotation files downloaded from GENCODE (Yin et al., 2020) (File S1). After the raw expression data were normalized, the limma package (Ritchie et al., 2015) was used to screen differentially expressed genes (DEGs, i.e., differentially expressed lncRNAs (DELs) and differentially expressed mRNAs (DEMs)) between samples isolated from elderly and middle-aged groups. The screening criteria were |log2 (fold change)| >1 and adjusted P < 0.05.

lncRNA-miRNA-mRNA triples prediction and functional enrichment analysis

The target miRNAs of DELs were collected using a putative microRNA target database miRcode (Jeggari, Marks & Larsson, 2012) and the results were used to predict downstream mRNAs. The interactions between miRNA and mRNA were supported by at least two of the following databases–miRDB (Wong & Wang, 2015), TargetScan 7.0 (Agarwal et al., 2015) and miRTarBase 6.0 (Hsu et al., 2011). The intersection of mRNA targets and DEMs were used for further analysis. A DEL-miRNA-DEM network was built with Cytoscape 3.7.1 (Shannon et al., 2003) software. Nodes with a high node degree (>5 connections), which was one of the network topological features , were considered to be the significant nodes in the network (Han et al., 2004).

Using the clusterProfiler package (Yu et al., 2012), Gene Ontology (GO) enrichment analysis and Kyoto Encyclopedia of Genes and Genomes (KEGG) pathway analysis of the involved DEMs were performed to further explore the potential biological functions of lncRNAs. The thresholds of the functional categories were P < 0.05 and a Benjamini–Hochberg corrected p-value < 0.05.

Animals and cell cultivation

This study was approved by the Ethics Committee of Tongji University affiliated Stomatological Hospital (Approval sequence: [2019]-DW-015). Female C57BL/6 mice were purchased from the Model Animal Research Center of Nanjing University (China). On arrival, all mice were housed in stainless plastic cages and were maintained under controlled environmental conditions throughout the study: 21–23 °C, with a relative humidity of 60 ± 10%, 12:12 h light–dark cycles (lights on from 0700 to 1900 h), and with commercial diet and water available. Cages were changed twice per week. When the mice reached a set age (i.e., 6 weeks or 12 months), they received a single intraperitoneal injection of Zoletil in distilled water at a dose of 50 mg/kg and were sacrificed by cervical dislocation. We used six mice for each group as biological replicates. Primary BMSCs derived from mouse femurs were cultured in the above-described medium at 37 °C in a 5% CO2 incubator. When the cell density reached 70–80%, osteoblast differentiation was induced in osteogenic medium (OM) which contained 100 nM dexamethasone, 100 nM ascorbic acid and 10 mM β-glycerophosphate (Sigma-Aldrich, USA). The medium was refreshed every three days and cells were harvested at the indicated times following treatments. During the experiment and routine feeding, animals received humane care according to the criteria outlined in the Guide for the Care and Use of Laboratory Animals published by the National Institutes of Health (Anonymous, 2011). This study was approved by the Ethics Committee of Tongji University Affiliated Stomatological Hospital.

The 293T cells were purchased from the Cell Bank of China and cultured in Dulbecco’s Modified Eagle’s Medium supplemented with 10% (v/v) fetal bovine serum (Gibco Life Technologies, USA) and, 1% (v/v) penicillin/streptomycin (HyClone, USA).

Alizarin Red staining and Alkaline phosphatase staining

BMSCs maintained in osteogenic induction medium for seven and twenty-one days were stained with Alizarin Red (Nanjing Jiancheng Bioengineering Institute, China) and BCIP/NBT alkaline phosphatase staining kit (Beyotime, China), respectively, to detect the osteogenic capacity of mouse BMSCs. Quantitative analysis of ALP activity was determined by an Enhanced BCA Protein Assay Kit (Beyotime, China) combined with an Alkaline phosphatase assay kit (Jiancheng, Nanjing, China).

Oil Red O staining

BMSCs were cultured in adipogenic induction medium (Cyagen, USA) for twenty-eight days and stained with Oil Red O (Sigma-Aldrich, USA) according to the provided instructions to detect the adipogenic capacity of mouse BMSCs.

Vectors and transfection

The miR-129-5p inhibitor/mimic vector and Antisense oligonucleotide (ASO) -MALAT1 were designed and synthesized by Ribobio Co., Ltd. Cell transfection was conducted with the help of a transfection Kit (Ribobio, China) and in accordance with the manufacturer’s instructions. Twenty-four hours post-transfection, quantitative reverse transcription PCR (qRT-PCR) was performed to confirm transfection efficiency, and samples were collected for further investigation.

qRT-PCR

Total RNA was extracted from BMSCs with RNAiso Plus (Takara, Japan) as per the manufacturer’s instructions. 1,000 ng of total RNA was reverse-transcribed to complementary DNA using Takara PrimeScript RT reagent kit (Takara, Japan) and following the manufacturer’s protocol.

MiRNA were reverse-transcribed using the transcriptor first strand cDNA synthesis kit (Roche, German). Equal amounts of complementary DNA were added to a 10 µL reaction mixture using Universal SYBR Green Master (Roche, German) and were subsequently used for real-time PCR reactions on a LightCycler96 Instrument (Roche, German).

The primer sequences of alkaline phosphatase (Alp), runt-related transcription factor 2(Runx2), osteocalcin (Ocn), U6, osteopontin (Opn) and D-glyceraldehde-3-phosphate dehydrogenase (Gapdh) are listed in Table 1. The sequences of miR-129-5p and lncRNA MALAT1 were synthesized by Ribobio (Guangdong, China). U6 was used as an internal control for miRNA, while Gapdh was used as an internal control of mRNA and lncRNA. All experiments were repeated in triplicate, and the relative RNA expression rates were calculated using the 2−ΔΔCt method.

Table 1 A list of primers.

Gene and primer type	Primer sequences	
STAT1		
Forward primer	5′-TCACAGTGGTTCGAGCTTCAG-3′	
Reverse primer	5′-GCAAACGAGACATCATAGGCA-3′	
Alp		
Forward primer	5′-GCCCTCTCCAAGACATATA-3′	
Reverse primer	5′-CCATGATCACGTCGATATCC-3′	
Runx2		
Forward primer	5′-CAAAGCCAGAGTGGACCCTT-3′	
Reverse primer	5′-AGACTCATCCATTCTGCCGC-3′	
Ocn		
Forward primer	5′-CTGACCTCACAGATCCCAAGC-3′	
Reverse primer	5′-TGGTCTGATAGCTCGTCACAAG-3′	
Opn		
Forward primer	5′-GATCAGGACAACAACGGAAAGG-3′	
Reverse primer	5′-GCTGGCTTTGGAACTTGCTT-3′	
Gapdh		
Forward primer	5′-AGGTCGGTGTGAACGGATTTG-3′	
Reverse primer	5′-TGTAGACCATGTAGTTGAGGTCA-3′	
U6		
Forward primer	5′-GCTTCGGCAGCACATATACTAAAAT-3′	
Reverse primer	5′-CGCTTCACGAATTTGCGTGTCAT-3′	

Western blotting

BMSCs were lysed in RIPA lysis buffer containing protease/phosphatase inhibiting cocktails (Beyotime, China). Total protein concentration was detected by the Enhanced BCA Protein Assay Kit (Beyotime, China). Equivalent amounts of protein lysates were loaded onto 12.5% SDS-PAGE gels (Beyotime, China) and then transferred to polyvinylidene fluoride membranes (Sangon Biotech, China). Membranes were blocked with 5% non-fat milk and hybridized with primary antibodies against signal transducer and activator of transcription 1 (STAT1) (1:1000; Abcam, UK), glyceraldehyde 3-phosphate dehydrogenase (GAPDH) (1:2000; BIOSS, China), lamin b1 (LAMIN B1) (1:10000; Abcam, UK) and RUNX2 (1:2000; Abcam, UK). GAPDH and LAMIN B1 were used as internal controls. The membrane was then hybridized with secondary antibodies for two hours. The blots were visualized and quantified using Odyssey CLx (LICOR, USA).

Dual-luciferase report assay

The lncRNA Malat1 and Stat1 sequences containing the predicted binding sites of miR-129-5p were amplified by PCR and inserted into the pmirGLO vector (Promega, USA). The resultant constructs were tagged as lncRNA Malat1-wild-type (lncRNA Malat1-wt)/ lncRNA Malat1-mutated-type (lncRNA Malat1-mut) and Stat1-wild-type (Stat1-wt)/ Stat1-mutated-type (Stat1-mut). The Malat1-mut and Stat1-mut were generated by site-directed mutagenesis, replacing the first six ribonucleotides of the miR-129-5p complementary sequence.

The 293T cells were seeded into 6-well plates (1.0 × 105) and co-transfected with miR-129-5p negative control (NC)/mimic and Malat1-wt/mut or Stat1-wt/mut using Lipofectamine 3000 (Invitrogen, USA) following the manufacturer’s instructions. Twenty-four hours after co-transfection at 37 °C, cells were harvested for luciferase assay using the dual-luciferase reporter (DLR) Assay System (Promega, USA). Transfection was performed three times.

Nuclear/cytoplasmic RNA and Nuclear/cytoplasmic protein extraction

Nuclear, cytoplasmic and total RNA were extracted by the PARIS™ Kit (Life technologies, USA). Simply put, cells were harvested and rinsed with phosphate buffered saline, and then centrifuged at 1,2000 rpm for five minutes. The cell sediments were mixed with 300 ul of cell fractionation buffer. Following centrifugation, cells were placed on ice for ten minutes. The solutions were then centrifuged at 500 g for five minutes (at 4 °C) and the supernatants containing cytoplasmic lysate were collected. The remaining sediments which contained nuclear protein/RNA were collected and mixed with 300 ul of cell disruption buffer and placed on ice. The cytoplasmic lysate and nuclear lysate fractions were used for RNA extraction, with the remainder being used for protein extraction. The lysis buffer used for RNA extraction was mixed with 300 µl of 2X lysis/binding solution and the same volume of ethanol. After being centrifugated, the mixture was transferred into a filter cartridge/collection tube assembly. Following three washing steps, the RNA was finally eluted with Elution Solution which was preheated to 95–100 °C. The ratio of nuclear gene expression to cytoplasmic expression of each gene is used for comparison (2−ΔCt method) (Khudayberdiev et al., 2013).

Micro-computed tomography(micro-CT)

Bone mass at the distal femur was assessed using micro-computed tomography (micro-CT) imaging (µCT50, Scanco Medical). After reconstructing image slices, the region of interest was manually selected in the marrow cavity. Ultra-high-resolution images (18 µm) of the specimens were obtained and relevant bone morphometric parameters, including BMD (mg/cc) and bone volume relative to total volume (BV/TV, %), were assessed.

Immunofluorescence staining

The cells were fixed with 4% Paraformaldehyde solution. Rabbit anti-mouse RUNX2 (Affinity Biosciences, OH, USA) was diluted 1:100 in PBS, and hybridized overnight at 4 °C. The samples were subsequently loaded with donkey anti-rabbit IgG immunofluorescence secondary antibodies (Beyotime, Shanghai, China) at 1:1000 dilution in PBS. The samples were also mounted with DAPI (1:10000, Sigma, US). Images were obtained under a confocal microscope (Leica Microsystems, Geremany).

Fluorescence in situ hybridization (FISH) assay

Fluorescence-labeled probes for lncRNA Malat1, 18S rRNA, and U6 RNA were designed and synthesized by Ribobio Co., Ltd. Fluorescence in situ hybridization staining was performed using a Ribo™ Fluorescent in situ hybridization Kit (RiboBio, Guangzhou, China). Images were acquired on an ECLIPSE Ts2R laser-scanning confocal microscope (Nikon, Japan).

RNA immunoprecipitation (RIP)

RIP was carried out by an EZ-Magna RIP™ RNA-Binding Protein Immunoprecipitation Kit (Millipore, USA) in accordance with manufacturer guidelines. BMSCs were lysed using RIP lysis buffer and aliquoted for analysis. 100 µL of the lysate was mixed with RIP buffer containing beads conjugated with Argonaute-2 (Ago2) (Abcam, UK) antibody and IgG (Abcam, UK) antibody, followed by the addition of proteinase K buffer. The ensuing protein samples and purified target RNA were used for WB and qRT-PCR assays, respectively.

Statistical analysis

Statistical analyses were performed using GraphPad Prism 8.0 software. Unless indicated otherwise, the data are presented as means ± standard deviations (SDs). Student’s t tests were used to compare results, with P < 0.05 indicating a statistically significant difference. One-way analysis of variance was used for multiple comparisons.

Results

Data quality control

We first conducted a data quality assessment. Regression analysis of raw data was performed using the affyPLM package, which provided us with relative log expression (RLE) boxplots, normalized unscaled standard errors (NUSE) boxplots, and RNA degradation plots (Heber & Sick, 2006). The RLE boxplot reflects the consistent trend of gene expression, which is defined as the logarithm of the expression value of a probe set in a sample divided by the median expression value of the probe set in all samples. The RLE plot revealed that the levels of gene expression in GSE35955 were relatively consistent, and the median value was close to 0 (Fig. 1A). This indicates that the sample quality of the chip data was reliable. NUSE is a more sensitive quality control method than RLE. NUSE is defined as the standard deviation of the perfect-match value of a probe set in a sample divided by the median standard deviation of the perfect-match value of the probe set in each sample. Consistent with the results of RLE, the nine samples’ NUSE values were approximately one, indicating their standard deviations were very close (Fig. 1B). Next, we normalized the data and annotated the probe names. A total of 16,418 mRNAs and 1,205 lncRNAs were identified in the microarray data through the human comprehensive gene annotation file (File S1).

Figure 1 Data quality assessment of the raw data sets.

(A) Boxplot representation of the relative log expression (RLE). (B) Boxplot representation of the normalized unscaled standard errors (NUSE).

The screening results of DELs and DEMs

To screen for genes that might function in BMSCs, we performed a differentially expressed gene analysis using the limma package (Ritchie et al., 2015). The screening criteria were |log2 (fold change)| >1 and adjusted P < 0.05. As a result, 64 differentially expressed lncRNAs (DELs) and 651 differentially expressed mRNAs (DEMs) were identified between the samples isolated from middle-aged and elderly groups (File S2). The expression of these DELs and all DEGs are visualized in a heatmap (Fig. 2) and a volcano plot (Fig. 3), respectively. MiRNAs associated with DELs were predicted through the miRcode database. At the same time, the mRNA targets of these miRNAs were predicted through three databases (i.e., miRTarBase (Hsu et al., 2011), TargetScan (Agarwal et al., 2015) and miRDB (Wong & Wang, 2015)) and were intersected with the previously obtained DEMs. Consequently, a total of 706 reliable miRNA-mRNA pairs and 347 predicted lncRNA-miRNA pairs (including 28 lncRNAs, 50 miRNAs and 281 mRNAs) were obtained for further analysis (File S3).

Figure 2 Heatmap of differentially expressed lncRNAs between BMSCs from middle-aged and old donors.

The horizontal axis shows the names of nine samples. The vertical axis presents the gene names.

Figure 3 Volcano plot of all differentially expressed genes between BMSCs from middle-aged and old donors.

FC are fold-change. Downregulated genes are green and upregulated genes are red.

Construction of the ceRNA network and Functional enrichment analysis

Using Cytoscape software (Shannon et al., 2003), a DEL-miRNA-DEM network was built based on the miRNA-DEM and DEL-miRNA pairs (Fig. 4A). The network consisted of 28 DEL nodes, 50 miRNA nodes and 281 DEM nodes.

Figure 4 The lncRNA associated ceRNA network and barplots of function enrichment analyses.

(A) The lncRNA-miRNA-mRNA ceRNA network. The parallelograms represent lncRNAs, the ellipses represent mRNAs, and the triangles represent miRNAs. (B) The top 10 most significant Gene ontology terms. (C) The top 10 most significant pathway terms.

Functional enrichment analyses based on DEMs, including GO analysis and KEGG analysis, may provide an approach to understanding the potential functions of DELs. We constructed a GO interaction network using the BiNGO tool (File S3). It uncovered ten significantly enriched terms (Fig. 4B and File S4), the top three of which were protein binding, focal adhesion and positive regulation of transcription from RNA polymerase II promoter. Twelve pathways were obtained from KEGG, as partially shown in Fig. 4C, including the terms of the AMPK, FoxO, and TGF-β signaling pathways (File S5). Interestingly, the AMPK (Wang et al., 2013; Zhou et al., 2019) and TGF-β signaling pathways (Zhang et al., 2017) were closely related to osteogenic differentiation of BMSCs.

Topological analysis of the ceRNA network

We computed the node degrees to identify the hub genes in the ceRNA network. In accordance with previous research (Han et al., 2004), we set the screening criterion as node degree ≥5. A total of 113 nodes were screened as hub genes, including 24 DELs, 49 miRNAs and 40 DEMs (Table 2). We found that lncRNA MALAT1 had higher node degrees, suggesting a potential role for BMSCs among elder donors.

Table 2 The list of differentially expressed genes(node degree > 5).

Number	Gene type	Gene name	Node degree	
1	miRNA	hsa-miR-17-5p	50	
2	miRNA	hsa-miR-129-5p	48	
3	miRNA	hsa-miR-24-3p	47	
4	miRNA	hsa-miR-3619-5p	40	
5	miRNA	hsa-miR-761	39	
6	miRNA	hsa-miR-429	39	
7	miRNA	hsa-miR-1297	38	
8	lncRNA	MALAT1	35	
9	miRNA	hsa-miR-20b-5p	33	
10	miRNA	hsa-miR-137	32	
11	miRNA	hsa-miR-206	31	
12	miRNA	hsa-miR-23b-3p	31	
13	miRNA	hsa-miR-613	30	
14	miRNA	hsa-miR-363-3p	27	
15	miRNA	hsa-miR-125b-5p	27	
16	miRNA	hsa-miR-125a-5p	26	
17	miRNA	hsa-miR-107	26	
18	miRNA	hsa-miR-27a-3p	26	
19	lncRNA	DIRC3	24	
20	miRNA	hsa-miR-217	24	
21	miRNA	hsa-miR-142-3p	23	
22	miRNA	hsa-miR-33a-3p	22	
23	miRNA	hsa-miR-507	22	
24	miRNA	hsa-miR-338-3p	22	
25	miRNA	hsa-miR-216b-5p	21	
26	miRNA	hsa-miR-135a-5p	20	
27	miRNA	hsa-miR-375	20	
28	lncRNA	AC008088	19	
29	lncRNA	COX10-AS1	18	
30	miRNA	hsa-miR-876-3p	18	
31	miRNA	hsa-miR-139-5p	17	
32	miRNA	hsa-miR-301b-3p	17	
33	lncRNA	AC010145	17	
34	lncRNA	RRN3P2	17	
35	lncRNA	LINC00520	16	
36	lncRNA	JPX	16	
37	lncRNA	AC017002	16	
38	lncRNA	C20orf197	16	
39	miRNA	hsa-miR-22-3p	16	
40	miRNA	hsa-miR-1244	16	
41	lncRNA	AP000462	15	
42	lncRNA	RUSC1-AS1	15	
43	miRNA	hsa-miR-449c-5p	15	
44	miRNA	hsa-miR-490-3p	15	
45	miRNA	hsa-miR-508-3p	15	
46	lncRNA	AC106801	14	
47	lncRNA	CECR3	14	
48	miRNA	hsa-miR-193a-3p	14	
49	miRNA	hsa-miR-140-5p	14	
50	miRNA	hsa-miR-212-3p	14	
51	lncRNA	LINC00269	13	
52	lncRNA	C22orf24	13	
53	miRNA	hsa-miR-590-5p	12	
54	miRNA	hsa-miR-10a-5p	12	
55	lncRNA	TDRG1	11	
56	miRNA	hsa-miR-455-5p	11	
57	miRNA	hsa-miR-4262	11	
58	lncRNA	ITPKB-IT1	10	
59	miRNA	hsa-miR-146b-5p	10	
60	mRNA	BACH2	9	
61	miRNA	hsa-miR-425-5p	9	
62	miRNA	hsa-miR-4465	9	
63	mRNA	SOWAHC	8	
64	mRNA	KCTD15	8	
65	mRNA	CNN3	8	
66	mRNA	PRDM1	8	
67	lncRNA	AP000347	8	
68	miRNA	hsa-miR-4500	8	
69	miRNA	hsa-miR-4458	8	
70	miRNA	hsa-miR-518a-3p	8	
71	mRNA	KIAA0232	7	
72	mRNA	NXT2	7	
73	mRNA	ENPP4	7	
74	mRNA	FLOT2	7	
75	mRNA	CDC42SE1	7	
76	mRNA	MKNK2	7	
77	mRNA	KLF13	7	
78	mRNA	TMEM170B	7	
79	lncRNA	AC073321	7	
80	lncRNA	LINC00208	7	
81	mRNA	HIPK1	6	
82	mRNA	E2F5	6	
83	mRNA	PDPK1	6	
84	mRNA	DDIT4	6	
85	mRNA	ETV1	6	
86	mRNA	ANKRD12	6	
87	mRNA	CAPRIN2	6	
88	mRNA	UHMK1	6	
89	mRNA	S1PR1	6	
90	mRNA	BNIP3L	6	
91	mRNA	JMY	6	
92	mRNA	NBEA	6	
93	mRNA	STX16	6	
94	mRNA	STRN3	6	
95	mRNA	VLDLR	6	
96	lncRNA	KCNQ1-AS1	6	
97	miRNA	hsa-miR-4735-3p	6	
98	mRNA	GPC4	5	
99	mRNA	FZD3	5	
100	mRNA	ZNF516	5	
101	mRNA	PKIA	5	
102	mRNA	TULP4	5	
103	mRNA	TAF1D	5	
104	mRNA	PPP3R1	5	
105	mRNA	ZNF281	5	
106	mRNA	EP300	5	
107	mRNA	SKI	5	
108	mRNA	MET	5	
109	mRNA	ATP6V1A	5	
110	lncRNA	DSCR10	5	
111	miRNA	hsa-miR-3666	5	
112	miRNA	hsa-miR-4295	5	
113	lncRNA	AL022341	5	

lncRNA Malat1 was upregulated but its target miR-129-5p was downregulated in BMSCs from old mice compared to young ones

To verify the predicted results of the bioinformatics analysis, we isolated the femurs of young (6 weeks) and old mice (12 weeks). The micro-CT images of the femurs indicated that bone mass was lower in old than young mice (Fig. 5A). The old mice consistently displayed a lower bone volume per tissue volume (BV/TV) (Fig. 5B). We further extracted BMSCs from mice femurs, which have a long filamentous structure (Fig. 5C). Alizarin Red and Oil Red O staining indicated that the BMSCs were able to differentiate into osteoblasts and adipocytes under corresponding induction conditions (Fig. 5C). Moreover, we found that the expression of Malat1 was significantly up-regulated in the BMSCs derived from femurs of aged mice (Fig. 5D), suggesting that Malat1 may play an important role. Based on previously obtained lncRNA-miRNA pairs, we also examined the expression of miRNAs with potential binding sequences to Malat1. As a result, miR-124a-5p, miR-125b-5p, miR-20b-5p, and miR-24-3p were actually upregulated in aging mice-derived BMSCs, with only miR-17-5p and miR-129-5p showing opposite trends (Fig. 5D). As mentioned above, ceRNAs are capable of reducing the abundance of miRNAs, so these two down-regulated genes were the potential targets of lncRNA Malat1. However, we found no apparent binding between Malat1 and miR-17-5p (File S6). Thus, we decided to explore the potential relationship between Malat1 and miR-129-5p. According to the obtained lncRNA-miRNA-mRNA triples (File S3), Malat1/miR-129-5p/Stat1 caught our attention. Stat1 is regulated by miR-129-5p and involved in BMSCs osteoblast differentiation (Xiao et al., 2016). Moreover, our previous study also indicated that Stat1 is an important transcription factor, which might contribute to the aging-related changes in BMSCs (Wu et al., 2019).

Figure 5 The expression of LncRNA Malat1 in BMSCs from young and aged mice.

(A)Representative micro-CT images showing the midshaft architectures of femurs from young and aged mice. N = 3. Scale bar, 1 mm. (B) Bone histomorphometric analysis of BMD and BV/TV in femurs from young and aged mice. N = 3. The mean ± s.e.m. is shown. (C) BMSCs showed a typical cobblestone-like morphology (a). Differentiation potential of BMSCs assessed by Alizarin Red staining (b) and Oil Red O (c). (D) The expression blot of lncRNA Malat1 and targets miRNA. At least three animals in each group were analysed, each experiment was repeated twice, and representative images are shown.

To determine the subcellular localization of Malat1, we performed a FISH assay and cytoplasmic/nuclear fractionation with BMSCs. Surprisingly, the amount of Malat1 observed in the nucleus was higher than in the cytoplasm (Figs. 6A and 6B). However, several studies (Huang et al., 2020; Yang et al., 2019b; Li, 2022) have proved that Malat1 had a potential ceRNA regulatory mechanism in BMSCs, so Malat1 expressed in the nucleus might regulate the biological function of BMSCs through other mechanisms.

Figure 6 LncRNA Malat1 was primarily localized in the nucleus.

(A) FISH analysis of Malat1 in BMSCs (The nuclei were stained with DAPI, and 18S rRNA was used as a cytoplasmic marker). (B) Cell nuclear/cytoplasmic fractionation and qRT-PCR showed the cellular districution of lncRNA Malat1 in BMSCs (U6 and Gapdh were used as separation quality standards and endogenous controls). Three independent experiments were performed for qRT-PCR assays.

lncRNA Malat1 was upregulated during osteoblast differentiation of BMSCs

Many important regulatory factors are involved in the process of osteoblast differentiation of BMSCs. Runx2 is essential for osteoprogenitor cell development and is an early transcription factor that determines the osteoblast differentiation of BMSCs (Kawane et al., 2018; Komor, 2010). When osteoprogenitor cells differentiate into pre-osteoblasts, the intracellular alkaline phosphatase (ALP) activity increases significantly, which is another marker for the early stage of osteoblast differentiation (Aubin, 2001). ALP can promote bone mineralization, and reflect the degree of differentiation and functional status of osteoblasts (Capulli, Paone & Rucci, 2014). With the further maturation of pre-osteoblasts, cells begin to produce osteopontin (OPN), bone sialoprotein (BSP), osteocalcin (OCN) and type 1 collagen, which are essential for bone formation and mineralization (Zoch, Clemens & Riddle, 2016). Among them, OCN is a hormone-like polypeptide produced and secreted by osteoblasts, which is considered as a marker for late-stage osteoblastogenesis (Nakashima & De Crombrugghe, 2003). To further understand the role of lncRNA Malat1 in osteoblast differentiation of BMSCs, we examined the dynamic changes of gene expression during the osteogenic differentiation of BMSCs. We found that Malat1 expression gradually increased over time (Fig. 7A). The dynamic expression of osteoblastic markers Runx2, Alp and Ocn during osteogenesis are shown in Figs. 7B–7D. The expression of Alp, an early osteogenic indicator, reached the highest peak after seven days of osteogenic induction, and then gradually decreased. However, another early osteogenic marker, Runx2, continued to show an upregulated trend after seven days. At the same time, the expression level of Ocn, an indicator of late-stage osteogenesis, increased with induction time. In view of the positive correlation between Malat1 expression and the expression of osteoblastic markers of BMSCs, we speculated that Malat1 may promote the osteogenic differentiation of BMSCs.

Figure 7 Expression patterns of lncRNA Malat1 during osteoblast differentiation of BMSCs.

(A) qRT-PCR analysis was used to detect the expression of lncRNA Malat1 during osteoblast differentiation of BMSCs at days 0,3,7,14 and 21. RNA expression at the indicated time points was normalized to day 0. Gapdh was used as an internal control. (B) qRT-PCR detection shows the expression of osteoblastic marker Runx2 on selected days. Gapdh was used as internal control. (C) qRT-PCR detection shows the expression of osteoblastic marker Ocn on selected days. Gapdh was used as internal control. (D) qRT-PCR detection shows the expression of osteoblastic marker Alp on selected days. Gapdh was used as internal control. Data are presented as means ± SD. Asterisk indicates P < 0.05, double asterisks indicate P < 0.01, triple asterisks indicate P < 0.001, quadruple asterisks indicate P < 0.0001 vs. control. Runx2, runt-related transcription factor 2; Ocn, osteocalcin; Alp, alkaline phosphatase. Three independent experiments were performed for qRT-PCR assays.

Inhibition of Malat1 suppressed osteogenic differentiation of BMSCs

To determine whether Malat1 expression played a role in BMSCs osteogenic differentiation, ASO against Malat1 was constructed and transfected into BMSCs. Compared with non-transfected control cells and the NC vector group, Malat1 expression was significantly downregulated in the ASO group (Fig. 8A), while the expression of miR-129-5p was upregulated (Fig. 8B). As a target gene of miR-129-5p, the expression level of Stat1 decreased after the disturbance of Malat1 expression (Fig. 8C). The results of qRT-PCR also showed that ASO-Malat1 also inhibited the expression of osteoblastic markers Alp, Runx2 and Ocn (Figs. 8D–8G). Consistent with these findings, the WB assay results indicated that the expression of RUNX2 and STAT1 in BMSCs decreased after treatment with ASO-Malat1 (Figs. 8H–8I). These results preliminarily demonstrated that Malat1 is indispensable for the osteogenic differentiation function of BMSCs. Each group of cells was then cultured in osteogenic medium (OM) or general medium (GM) for seven days. Osteogenic induction significantly enhanced the activity of the ALP enzyme, while inhibition of lncRNA Malat1 suppressed this effect (Figs. 8J–8K), suggesting that Malat1 played an important role in the early stage of osteogenic differentiation of BMSCs. Immunofluorescence staining showed that ASO-Malat1 reduced the expression of RUNX2 protein in BMSCs (Fig. 9A). As a transcription factor, Runx2 regulates the transcriptional activity by binding to the promoters or enhancers of the target genes, so the process of Runx2 translocation into the nucleus is crucial for osteoblast differentiation (Nakashima et al., 2002). In that case, we also extracted the nuclear/cytoplasmic protein to detect the expression level of RUNX2 in different regions. The content of RUNX2 protein in the nucleus as well as the cytoplasm was reduced, which further supports the effect of Malat1 on osteogenic differentiation (Figs. 9B–9C). Similar to RUNX2, STAT1 translocates into the nucleus after being activated in the cytoplasm, and regulates the expression of target genes, including nuclear factor-kappa B, myelocytomatosis oncogene, and matrix metallopeptidase 9 (Görlich et al., 1995; ten Hoeve et al., 2002). As shown in Fig. 9D, the total content of STAT1 in BMSCs was downregulated in BMSCs after treated with ASO. The protein expression of intranuclear STAT1 decreased in the ASO group, while the expression of intracytoplasmic STAT1 increased. This may result from reduced activated STAT1 or blocked STAT1 nuclear translocation. These results suggest that Malat1 supports osteoblast differentiation of BMSCs, but whether this effect is related to miR-129-5p and Stat1 remains unclear.

Figure 8 Inhibition of lncRNA Malat1 suppressed osteogenic differentiation of BMSCs.

(A) qRT-PCR detection shows expression of lncRNA Malat1 in BMSCs after transfection with siRNA against lncRNA Malat1 (ASO) or the negative control (NC) vector for 24 h. Gapdh was used as an internal control. Data are presented as means ± SD. (B) qRT-PCR detection shows expression of miR-129-5p in BMSCs after transfection with siRNA against lncRNA Malat1 (ASO) or the negative control (NC) vector for 24 h. U6 was used as an internal control. Data are presented as means ± SD. Relative mRNA expression of Stat1 (C), Alp (D), Runx2 (E), Ocn (F) and Opn (G) was measured by qRT-PCR at day 7 of OM induction. Gapdh was used for normalization. Data are presented as means ± SD. (H) The expression of STAT1 and RUNX2 was detected by Western blot analysis when lncRNA Malat1 was knocked down in BMSCs. (I) Protein expression of STAT1 and RUNX2 in BMSCs transfected with ASO or NC vector. Asterisk indicates P < 0.05, double asterisks indicate P < 0.01 vs. Blank. (J) Images of ALP staining in BMSCs after culture in GM or OM for 7 days. Three independent experiments were performed for all of the blots and qPCR assays, and representative images are shown. (K) Histograms show ALP activity by spectrophotometry. Data are presented as means ± SD.

Figure 9 Inhibition of lncRNA Malat1 impeded the nuclear translocation in Runx2.

(A) Immunofluorescence staining was applied to test the effects of siRNA against lncRNA Malat1 (ASO) on the protein levels of RUNX2. (B) Western blot assay was conducted to test protein levels of total, nuclear, cytoplasmic RUNX2 in BMSCs transfected with ASO or NC vector. Three independent experiments were performed, and representative images are shown. (C) Quantitative analysis of protein expression of RUNX2 in BMSCs transfected with ASO or NC vector. Double asterisks indicate P < 0.01, triple asterisks indicate P < 0.001, quadruple asterisks indicate P < 0.0001 vs. blank group. (D) Quantitative analysis of protein expression of RUNX2 in BMSCs transfected with ASO or NC vector. An asterisk (*) indicates P < 0.05 vs. blank group.

Figure 10 Inhibition of miR-129-5p promoted differentiation of BMSCs.

(A) qRT-PCR detection shows expression of miR-129-5p in BMSCs after transfection with miR-129-5p inhibitors or NC for 24 h. Gapdh was used as an internal control. Data are presented as means ± SD. (B) qRT-PCR detection shows expression of Malat1 in BMSCs after transfection with miR-129-5p inhibitors or NC for 24 h. U6 was used as an internal control. Data are presented as means ± SD. Relative mRNA expression of Stat1 (C), Alp (D), Runx2 (E), Ocn (F) and Opn (G) was measured by qRT-PCR at day 7 of OM induction. Gapdh was used for normalization. Data are presented as means ± SD. (H) The expression of STAT1 and RUNX2 was detected by Western blot analysis when miR-129-5p was knocked down in BMSCs. (I) Protein expression of STAT1 and RUNX2 in BMSCs transfected with miR-129-5p inhibitor or NC vector. An asterisk (*) indicates P < 0.05, double asterisks indicate P < 0.01 vs. control. (J) Images of ALP staining in BMSCs after culture in GM or OM for 7 days. Three independent experiments were performed for all of the blots and qPCR assays, and representative images are shown. (K) Histograms show ALP activity by spectrophotometry. Data are presented as means ± SD.

Figure 11 Inhibition of miR-129-5p did not directly affect the nuclear translocation in Runx2.

(A) Immunofluorescence staining was applied to test the effects of miR-129-5p inhibitor on the protein levels of Runx2. (B) Western blot assay was conducted to test protein levels of total, nuclear, cytoplasmic RUNX2 in BMSCs transfected with miR-129-5p inhibitor or NC vector. Three independent experiments were performed, and representative images are shown.

Inhibition of miR-129-5p promoted osteogenic differentiation of BMSCs

MiRNAs play an important role in mesenchymal stem cells. For example, overexpression of miR-485-5p can induce a senescence-like phenotype and proliferation inhibition of adipose tissue-derived mesenchymal stem cells (Kim et al., 2012). Another research indicated that age-related bone loss can be rescued by regulating miR-142-3p/Bmal1/YAP signaling axis in BMSCs (Cha et al., 2022). To investigate the role of miR-129-5p in the osteogenic differentiation of BMSCs and its underlying mechanism, a miR-129-5p inhibitor was constructed and transfected into BMSCs to determine the effect of miR-129-5p on the osteogenic differentiation of BMSCs. The results showed an acceptable inhibition effect (Fig. 10A). In addition, with the inhibition of miR-129-5p, the relative mRNA level of Malat1 and Stat1 were significantly upregulated (Figs. 10B–10C). Similarly, Runx2, Opn and Ocn expression were also upregulated by miR-129-5p inhibition (Figs. 10D–10G), suggesting that the effect of miR-129-5p on the BMSC osteoblast differentiation is opposite to that of Malat1. The western blot results indicated that the protein amount of both STAT1 and RUNX2 increased in the experimental group (Figs. 10H–10I). The ALP enzyme activity of BMSCs increased after osteogenic induction, and inhibition of miR-129-5p further enhanced this effect (Figs. 10J–10K). This suggests that miR-129-5p has the function of promoting the osteogenic differentiation of BMSCs. The results of immunofluorescence staining were similar to those of WB, and miR-129-5p inhibitor significantly up-regulated the expression of RUNX2 (Fig. 11A). To clarify whether miR-129-5p regulates the osteogenic differentiation of BMSCs by affecting the nuclear translocation of RUNX2, we also performed nucleocytoplasmic separation experiments. The results showed that the content of RUNX2 in the nucleus increased significantly, while the cytoplasmic content decreased (Figs. 11B–11C), suggesting that miR-129-5p may affect the osteogenic differentiation of BMSCs through Runx2 signaling pathways. The result of nuclear/cytoplasmic protein extraction assay also showed that the protein expression of intracytoplasmic STAT1 increased in the miR-129-5p inhibitor treated group (Fig. 11D). However, the expression level of nuclear STAT1 in the miR-129-5p inhibitor group showed no significant difference. These results suggest that miR-129-5p inhibit osteoblast differentiation of BMSCs, but whether this effect is related to Stat1 still needs further research.

miR-129-5p is a potential target of lncRNA Malat1

Most miRNAs directly bind to the target gene sequence through the RNA-induced silencing complex. Therefore, we used an online database (http://starbase.sysu.edu.cn/) to predict the potential binding site between miR-129-5p and Malat1, finding that Malat1 has a broadly conserved binding site with miR-129-5p. Next, we constructed an original version of lncRNA Malat1 and a mutated one with seven altered complementary nucleotides (Fig. 12A). The luciferase reporter gene plasmids containing the two sequences were co-transfected into 293T cells with miR-129-5p mimics or scramble mimics. The results showed that miR-129-5p mimics could significantly downregulate the luciferase activity of the wild-type Malat1 groups (Fig. 12B). However, this phenomenon was not observed in the two Malat1-mutation groups, indicating that miR-129-5p specifically targets lncRNA Malat1. The result of the qRT-PCR assay further confirmed that Malat1 inhibition significantly enhanced miR-129-5p expression in BMSCs (Fig. 12C).

Figure 12 miR-129-5p is a potential target of lncRNA Malat1.

(A) Complementary bases between the sequences are labeled with red font. The sequence of the mutant lncRNA Malat1 construct is also shown as underlined. (B) Dual-luciferase reporter assay of 293T cells co-transfected with lncRNA Malat1-wt, or lncRNA Malat1-mut and with miR-129-5p mimic or NC. (C) MiR-129-5p expression in BMSCs was measured by qRT-PCR following transfection with ASO for 24 h. Data are presented as means ±  SD. Two independent experiments were performed.

Potential association of Stat1 with miR-129-5p

Although we found that interfering with miR-129-5p did not affect the expression level of Stat1 in BMSCs, we still examined the binding relationship between the two genes, to potentially verify the results of a previous study (Xiao et al., 2016). Following the previous study’s methods (Xiao et al., 2016), we assessed the possible interaction between miR-129-5p and Stat1, by first predicting the potential binding site of Stat1 using the Starbase database. However, the binding site was in the coding sequence (CDS) instead of the 3′UTR region. As described above, luciferase reporter gene plasmids containing two versions of Stat1 were constructed and co-transfected into 293T cells along with miR-129-5p mimics or scramble mimics (Fig. 13A). It appeared that miR-129-5p mimics also downregulated the luciferase activity of the wild-type Stat1 groups, while this phenomenon was not observed in both Stat1-mutation groups (Fig. 13B). However, the degree of downregulation of luciferase activity in 293T cells in the wild-type Stat1 group after transfected with miR-129-5p was different from the previous study (Xiao et al., 2016). Conversely, qRT-PCR analyses indicated that overexpression of miR-129-5p suppressed Stat1 expression in BMSCs (Fig. 13C), while the protein level of STAT1 showed no significant difference (Fig. 13D). Overall, it is still uncertain whether Stat1 is a direct target of miR-129-5p.

Figure 13 Potential association of Stat1 with miR-129-5p..

(A) Sequence alignment between miR-129-5p and Stat1. (B) Dual-luciferase reporter assay of 293T cells co-transfected with Stat1-wt or Stat1-mut and miR-129-5p mimic or NC. QRT-PCR (C) and western blot (D) detection show the expression of Stat1 in BMSCs following transfection with miR-129-5p mimic or NC for 24 h. Data are presented as means ± SD. Two independent experiments were performed.

MiR-129-5p inhibitor reversed the inhibitory effect of ASO-Malat1 on osteogenic differentiation of BMSCs

To prove that the Malat1/miR-129-5p signal axis regulated the osteogenic differentiation ability of BMSCs through miR-129-5p, we co-transfected miR-129-5p mimics and ASO-Malat1 into BMSCs using osteogenic induction. The findings of the qRT-PCR assay indicated that inhibition of miR-129-5p reversed the inhibitory effect of ASO-Malat1 on osteogenic differentiation by promoting Alp, Opn, Ocn, and Runx2 expression (Figs. 14A–14D). Consistently, BMSCs transfected with only ASO-Malat1 showed a decreased level of ALP activity, while the addition of miR-129-5p reversed the inhibitory effect (Fig. 14E and 14F). MiRNA degrades mRNA or hinders its translation by forming a so-called RNA-induced silencing complex (RISC) (Klein et al., 2017). Ago2 is the core component of RISC, linking miRNAs and their mRNA target sites (Klein et al., 2017). Therefore, immunopurification of Ago2 under suitable conditions can obtain mutually bound miRNA and mRNA (Yang et al., 2019b). In the RIP quality control experiment, the anti-Ago2 group successfully pulled down some Ago2 protein (Fig. 15A). In contrast, the anti-IgG group had no obvious bands (Fig. 15A), suggesting that the quality of the RIP assay was reliable. The RIP assay showed that anti-Ago2, an important part of the RNA silencing inducing complex, significantly enriched Malat1 and miR-129-5p in contrast with anti-IgG, indicating an endogenous combination between miR-129-5p and lncRNA Malat1 (Fig. 15B).

Figure 14 Inhibition of miR-129-5p reversed the inhibitory effect of ASO on osteogenic differentiation of BMSCs.

(A–D) Relative mRNA expression of Alp (A), Runx2 (B), Ocn (C) and Opn (D) were measured by qRT-PCR at day 7 of OM induction. Gapdh was used for normalization. Data are presented as means ± SD. (E) Images of ALP staining in BMSCs transfected with ASO or miR-129-5p inhibitor after culture in OM for 7 days. (F) Histograms show ALP activity by spectrophotometry. Data are presented as means ± SD. Three independent experiments were performed.

Figure 15 lncRNA-Malat1 sponges with miR-129-5p.

(A) RIP assay shows that the miRNAs were successfully pulled-down using an Ago2 antibody. IgG was used as the negative control. (B) RNA levels of lncRNA Malat1 and miR-129-5p after immunoprecipitation were determined by qRT-PCR. Gapdh was the reference gene of lncRNA Malat1. U6 was used as the reference gene of miR-129-5p. Numbers are mean ± SD. Three independent experiments were performed.

Discussion

The age-related degradation of bone microstructure can lead to fragility fractures, and BMSCs play a critical role in bone mineralization changes (Ensrud & Crandall, 2017). The proliferation and effector functions of BMSCs are regulated by intracellular and extracellular stimuli, including exosomes, metabolites, and lncRNAs, some of which are related to aging (Concha et al., 2019; Jiang et al., 2021). Therefore, we screened differentially expressed lncRNAs based on the expression profile of BMSCs from elderly and young donors, finding that lncRNA Malat1 expression was significantly increased in the elderly group.

Several studies have investigated how lncRNA Malat1 regulates the osteogenic differentiation of BMSCs through the ceRNA mechanism. Huang et al. (2020) found that Malat1 could promote the osteogenic differentiation of BMSCs by sponging miR-214. Yang et al. (2019b) determined that Malat1 functioned as a sponge for miR-34c, and thus enhanced the osteoblast activity of BMSCs. Moreover, Malat1 was also capable of sponging miR-124-3p to exert the same effect in BMSCs (Li, 2022). Interestingly, another study found that Malat1 can inhibit the alkaline phosphatase activity of BMSCs by activating the MAPK signaling pathway (Zheng et al., 2019). These results suggest that Malat1 has a dual-effect on the osteogenic differentiation of BMSCs, and its mechanism may be far more complicated. In this current study, the osteogenic differentiation ability of BMSCs was suppressed by ASO-Malat1, which could also inhibit the expression of Malat1. Given that the ceRNA mechanism has received increasing attention in recent years, we also sought to explain this inhibitory effect by searching for another lncRNA/miRNA signal axis. At first, we chose miR-17-5p and miR-129-5p as hub genes, both of which had binding sites for Malat1 and their expression was negatively correlated with Malat1 expression. However, DLR experiments indicated that only miR-129-5p interacts with Malat1. A recent study also found that MALAT1 can bind to miR-129-5p and downregulate its expression in pancreatic cancer cells (Xu et al., 2021). Additionally, the inhibition of BMSC osteogenic ability caused by a lack of Malat1 can be reversed by miR-129-5p inhibitors, suggesting that Malat1 plays a regulatory role through sponging miR-129-5p. It is worth noting that through the FISH experiment and the nuclear/cytosol fractionation experiment, we found that Malat1 expression was higher in the nucleus than cytoplasm, which contradicts the findings of a recent study (Li, 2022). Since the ceRNA regulation pattern of Malat1 has been confirmed by several studies, it is possible that these nuclear-expressed Malat1 may also regulate the osteogenic differentiation of BMSCs by affecting important transcription factors.

Research has suggested that miR-129-5p can negatively regulate Runt-related transcription factor 1 and subsequently inhibit the cartilage differentiation ability of BMSCs (Zhu et al., 2021). An earlier study found that miR-129-5p reduced the expression level of Stat1, resulting in a dramatic increase in RUNX2 protein levels in murine BMSCs (Xiao et al., 2016). Our study determined that overexpression of miR-129-5p inhibited the expression of Stat1 to a certain extent. However, the dual-luciferase reporter gene experiment and the RIP assay showed that the interaction between miR-129-5p and Stat1 is not obvious. Interestingly, the predicted bind site of Stat1 (located in the CDS of Stat1) was consistent with a previous study (Xiao et al., 2016). While several studies have pointed out that a large number of miRNA binding sites exist in the CDS region (Helwak et al., 2013; Brümmer & Hausser, 2014), few results were related to these binding sites. Zhang et al. (2018) discovered and confirmed a novel class of miRNA response elements that function in the CDS region. MiRNAs induce transient ribosome stalling to repress translation without affecting mRNA levels (Zhang et al., 2018). This discovery provides a new idea for improving the regulatory mechanism of miRNA.

In summary, we conducted in vitro experiments and found that lncRNA Malat1 could affect the osteogenic differentiation process of BMSCs by regulating miR-129-5p. However, further research is needed to determine the specific mechanism through which miR-129-5p affects osteogenic differentiation and applications for the use of lncRNA Malat1 in bone tissue engineering.

Supplemental Information

Supplemental Information 1 Data banks/repositories corresponding to all datasets analyzed in this study

Click here for additional data file.

Supplemental Information 2 All differentially expressed genes identified between young and old donors

Click here for additional data file.

Supplemental Information 3 Potential co-expression competing triples (lncRNA-miRNA-mRNA triples)

Click here for additional data file.

Supplemental Information 4 The enriched Gene ontology(GO) terms of linked mRNAs in the ceRNA network

Click here for additional data file.

Supplemental Information 5 The enriched pathway terms of linked mRNA in the ceRNA network

Click here for additional data file.

Supplemental Information 6 miR-17-5p is not a potential target of lncRNA Malat1

(A) Complementary bases between the sequences are labeled with red font. The sequence of the mutant lncRNA Malat1 construct is also shown as underlined.

(B) Dual-luciferase reporter assay of 293T cells co-transfected with lncRNA Malat1, or lncRNA Malat1-Mut and with miR-17-5p mimic or miR-NC.

Click here for additional data file.

Supplemental Information 7 Full-length uncropped blots

Click here for additional data file.

Supplemental Information 8 Raw data for Fig. 4 (Go Analysis)

Click here for additional data file.

Supplemental Information 9 Raw data for Fig. 4 (KEGG Analysis)

Click here for additional data file.

Supplemental Information 10 Raw data for Fig. 5 (PCR Analysis)

Click here for additional data file.

Supplemental Information 11 Raw data for Fig. 5 (microCT Analysis)

Click here for additional data file.

Supplemental Information 12 Raw data for Fig. 6 (PCR Analysis)

Click here for additional data file.

Supplemental Information 13 Raw data for Fig. 7 (PCR Analysis)

Click here for additional data file.

Supplemental Information 14 Raw data for Fig. 8 (PCR Analysis)

Click here for additional data file.

Supplemental Information 15 Raw data for Fig. 8 (AKP Activity)

Click here for additional data file.

Supplemental Information 16 Raw data for Fig. 10 (PCR Analysis)

Click here for additional data file.

Supplemental Information 17 Raw data for Fig. 12 (PCR Analysis)

Click here for additional data file.

Supplemental Information 18 Raw data for Fig. 12 (Dual luciferase)

Click here for additional data file.

Supplemental Information 19 Raw data for Fig. 13 (PCR Analysis)

Click here for additional data file.

Supplemental Information 20 Raw data for Fig. 13 (Dual luciferase)

Click here for additional data file.

Supplemental Information 21 Raw data for Fig. 14 (PCR Analysis)

Click here for additional data file.

Supplemental Information 22 Raw data for Fig. 14 (AKP Activity)

Click here for additional data file.

Supplemental Information 23 Raw data for Fig. 15 (PCR Analysis)

Click here for additional data file.

The authors would like to express their gratitude to EditSprings for the expert linguistic services provided.

Abbreviations

Ago2 argonaute-2

Alp alkaline phosphatase

BiNGO Biological Networks Gene Ontology tool

BMSCs bone mesenchymal stem cells

ceRNA competitive endogenous RNA

DEG differentially expressed gene

DEL differentially expressed lncRNA

DEM differentially expressed mRNA

DLR dual-luciferase reporter

FISH fluorescence in situ hybridization

Gapdh D-glyceraldehde-3-phosphate dehydrogenase

GEO Gene Expression Omnibus database

GO Gene Ontology

KEGG Kyoto Encyclopedia of Genes and Genomes

lncRNA long non-coding RNA

Malat1 metastasis-associated lung adenocarcinoma transcript 1

micro-CT micro-computed tomography

miRNA microRNA

mRNA messenger RNAs

NCBI National Center for Biotechnology Information

ncRNA non-coding RNA

NUSE Normalized unscaled standard errors

Ocn osteocalcin

Opn osteopontin

RIP RNA binding protein immunoprecipitation

RLE relative log expression

Runx2 runt-related transcription factor 2

Additional Information and Declarations

Competing Interests

Author Contributions

Animal Ethics

Data Availability

The authors declare there are no competing interests.

Junhao Yin conceived and designed the experiments, performed the experiments, prepared figures and/or tables, authored or reviewed drafts of the paper, and approved the final draft.

Zhanglong Zheng conceived and designed the experiments, performed the experiments, analyzed the data, prepared figures and/or tables, and approved the final draft.

Xiaoli Zeng and Zexin Ai conceived and designed the experiments, analyzed the data, prepared figures and/or tables, and approved the final draft.

Yijie Zhao conceived and designed the experiments, analyzed the data, authored or reviewed drafts of the paper, and approved the final draft.

Miao Yu and Yang’ou Wu conceived and designed the experiments, prepared figures and/or tables, and approved the final draft.

Jirui Jiang performed the experiments, authored or reviewed drafts of the paper, and approved the final draft.

Jia Li and Shengjiao Li performed the experiments, prepared figures and/or tables, authored or reviewed drafts of the paper, literature review, and approved the final draft.

The following information was supplied relating to ethical approvals (i.e., approving body and any reference numbers):

Ethics Committee of Tongji University Affiliated Stomatological Hospital approved the research ([2019]-DW-015).

The following information was supplied regarding data availability:

The data banks/repositories corresponding to all datasets analyzed in this study and the raw data are available as Supplementary Files.

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
