# Peer review of "lncRNA MALAT1 mediates osteogenic differentiation of bone mesenchymal stem cells by sponging miR-129-5p"

_PeerJ, doi:10.7717/peerj.13355_

## Round 0.1 · original submission · Major Revisions

Please address all the concerns of all 3 reviewers.

Reviewer 1 ·

Basic reporting

The introduction is very succinct and does not provide sufficient background for a large audience. The authors should explain differentiation of BMSCs in more details and introduce the key molecular players. Please also introduce lncRNAs and miRNAs; what they are and what they do.
The overall flow can be also improved. Currently the manuscript resemble a bullet point list of results rather than a story and sometimes lacks logical progression. For example, it is not clear why authors focus on miR-129-5p or miR-17-5p (line 248). These miRNAs were not introduced in the background section. Is it because Malat1 has binding sites of miR-129-5p and miR-17-5p?
The authors often refer the reader to Figures without telling the reader what their interpretations are. For example, lines 263–265 read, “The dynamic expression of osteoblastic markers Runx2, ALP and OCN during osteogenesis are shown in Figures 7B-D.” What are these changes? Does expression of Runx2, ALP and OCN increase or decrease during osteogenesis? Are these changes concomitant with changes in Malat1 expression?
Figures should be re-arranged. There are currently 15 Figures, and some panels are not cited in a chronological order (e.g., Figure 10B is referred in the text before Figure 10A). Also, please consider using a different color palette to facilitate interpretation for color-blind readers.
Figures 8J and 10J: what are the round inserts? I suppose, they are pictures of cell plates. If this is the case, indicate which region of the plate is shown below at a higher magnification, Also, the scale bar is barely visible.
Figure 9A: Runx2 signal cannot be seen easily, even in the Control sample, while in Figure 11A the signal appears fine. Please enhance the contrast for more visibility.

Experimental design

Please provide the reader all key information in the Material and Methods or Results sections, so that the reader does not need to search for it. For example, lines 220–221 read, “lncRNA-miRNA pairs (including 28 lncRNAs, 50 miRNAs and 281 mRNAs) were obtained as previously described (8).” Also, explain non-common abbreviations. What are NUSE values (line 212)? One does not need to introduce an acronym for an expression used only once in the text.
It is unclear to me which groups are compared in the section 3.3. Line 217 mentions “the tumor and adjacent normal tissue”, but lines 240–241 state “We found that lncRNA MALAT1 had higher node degrees, suggesting a potential role for BMSCs among elder donors.” Are tumor and normal tissues compared, or are tissues from young and old donors compared?

Validity of the findings

The authors should quantify their Western Blots. I am not convinced that there is a reduction in Stat1 levels.
Avoid expressions such as “slightly reduced” (lines 278 and 289) and “the change in luciferase activity was not obvious” (line 312). Please quantify your results and provide statistics.
The authors report that Malat1 is localized in the nucleus, therefore I am confused about its role as a miRNA sponge. miRNAs mature and function in the cytoplasm, so how does Malat1 interact with miRNAs? The authors perform AGO2 pull-downs and find miR-129-5p, Malat1 and Stat1 in the pull-down fraction. Stat1 and miR-129-5p make sense but finding nuclear Malat1 associated with AGO2 is surprising. I believe that this is an artefact and miR-129-5p binds Malat1 after the cells are lyzed and thereby compartmentalization is abolished.
It is not clear to me how Stat1 luciferase assay was performed. Did the authors clone miR-129 binding sites from Stat1 CDS into Luciferase 3’ UTR? If this is the case, I do not think that it is a meaningful experiment as even if miR-129 binds these sites, this binding is likely abolished by ribosomes upon translation.

Additional comments

The authors performed a lot of work and carried out an overall thorough analysis. I believe that points that I mentioned above will improve the manuscript.

·

Basic reporting

The topic and content discussed in this manuscript are within the scope of the journal.

The English of your manuscript must be improved before resubmission. I strongly suggest that you obtain assistance from a colleague who is well-versed in English or whose native language is English.

Experimental design

The organization and subsections are also appropriate. The manuscript is structured and presented in a reader-friendly manner.
The survey methodology is consistent with comprehensive and unbiased coverage of the subject.

Validity of the findings

In this manuscript, there are well-developed and supported arguments that meet the goals set out in the Introduction, and the authors also provided very good descriptions and summaries of the important role of long noncoding RNA (lncRNA) metastasis-associated lung adenocarcinoma transcript 1 (Malat1). And the authors also indicated that lncRNA Malat1 may play a critical role in maintaining the osteoblast differentiation potential of BMSCs by sponging miR-129-5p.

The paper has minor, easily fixable, technical flaws that do not impact the validity of the main results

Additional comments

1. In line 217, I suggest using all Arabic numerals consistently
2. Please indicate how many repeats are used in each experiment.
3. In Figure 6B, the authors show Malat1 expression using qRT-PCR, I wonder if you use the internal control shown in Figure 7. I know that GAPDH amplification is not available in Nucleus. So, I don't know how the author got the result. I suggest the authors verify the expression of Malat1 by Western blot.
4. In Figure 7, consider validating the expression of these genes at the protein level.
5. In Figure 8A, I suggest the authors check knockdown Malat1 expression at the protein level. In Figure 8H, consider repeating 3 times in each group, and the quantitative analysis data of western blot analysis is missing, which should be presented.

Reviewer 3 ·

Basic reporting

The authors should do proofreading more carefully

Experimental design

Methods:

What is the gender for the donors?

The age and quantity of the mice in each group should be mentioned in 2.5

Results:

Why was the bioinformatic analysis done in middle aged and elderly donors but animal-based study conducted in young and old mice?

Were 64 DELs and 651 DEMs identified between middle aged and old donors or between the tumor and adjacent normal tissues as stated in line 217?

In figure 7, did the authors do any statistical analysis or did they forget to label p-value in all plots?

In figure 8, 10 and 14, other than the mRNA quantification for those BMSCs differentiation markers, the authors should also quantify their changes at protein level via western blot analysis.

In figure 8(j) and 10(j), the authors should label which pictures are culture in GM or OM and explain in the result part how these support their conclusion.

The authors should add a MiR129-5p inhibitor only group in figure 14 (A)-(D) and a mRNA expression analysis of Malat1. As suggested by the mRNA analysis in figure 10, the inhibition of MiR129-5p itself can promote expression of BMSCs differentiation and the data in (A)-(D) so far are not sufficient to indicate a reversal effect on ASO-Malat1.

In figure 14 (F), the authors should consider adding a ASO+OM group and redo this assay since it is obviously in conflict with what has been shown in the ALP staining of figure (E).

Please check all the labels in each picture and make sure they are consistent throughout the manuscript. For example, the MiR129-5p inhibitor was presented as "inhibitor" in figure 10 and "MiR 129 inhibitor" in figure 14 figure A-D and F but MiR129-5p inhibitor in figure E.

Validity of the findings

The validity of the findings in this study is largely limited by the small sample sizes in both bioinformatic (n=5, n=4) and animal study (n=3).

---

## Round 0.2 · accepted · Accept

This decision is based on recommendation of acceptance by two independent reviewers.

Reviewer 1 ·

Basic reporting

The authors addressed all of my comments.

Experimental design

N/A

Validity of the findings

N/A

Additional comments

N/A

·

Basic reporting

The organization and subsections are also appropriate.

Experimental design

The topic and content discussed in this manuscript are within the scope of the journal.

Validity of the findings

It is better if the quality of the western blotting picture is higher.

Additional comments

No